


# Laboratory and In-flight Evaluation of a Cloud Droplet Probe (CDP)

Spencer Faber[1], Jeffrey R. French[1], Robert Jackson[1,2]

[1]Department of Atmospheric Science, University of Wyoming, Laramie, WY, 82071, USA
[2]*Present Affiliation:* Argonne National Laboratory, Environmental Science Division, Argonne, IL, 60439, USA

*Correspondence to*: Jeffrey R. French (jfrench@uwyo.edu)

**Abstract.** Laboratory and in-flight evaluations of measurements from a Cloud Droplet Probe (CDP) are presented. A description of a water droplet-generating device, similar to those used in previous studies, is provided along with validation of droplet sizing and positioning. Laboratory evaluations of a CDP using the droplet generating system indicate errors in

sizing that depend on both droplet diameter and position within the sample area through which a droplet transited. For the smallest diameters tested, the CDP undersized droplets by 1 – 4 µm for the majority of those sampled. The remaining droplets were sized to within 1 µm of the actual diameter. Droplets with diameters of 17 and 24 µm were sized correctly, within 2 µm, which is the nominal CDP bin width for droplets of that size. For all larger diameters, the majority of droplets were oversized by 2 – 4 µm, while a small percentage were severely undersized, by as much as 30 µm. This combination

leads to an artificial broadening of the spectra, although errors in higher order moments were generally less than 10%. Comparisons of liquid water content (LWC) calculated from the CDP and that measured from a Nevzorov hotwire probe were conducted for 17,917 1 Hz in-cloud points. Although some differences were noted based on volume-weighted mean diameter and total droplet concentration, the CDP-estimated LWC exceeded that measured by the Nevzorov by approximately 20%, more than twice the expected difference based on results of the laboratory tests and considerations of

Nevzorov collection efficiency.

## 1 Introduction

In-situ cloud studies often utilize measurements from forward scattering optical particle counters (OPCs) to provide size and concentration information about cloud hydrometeors up to a few 10's of microns in diameter. The Particle Measuring Systems' Forward Scattering Spectrometer Probe (FSSP) and the Droplet Measurement Technologies' (DMT) Cloud

Droplet Probe (CDP) are forward scattering OPCs used to measure hydrometeors of 1 – 50 µm in diameter. These instruments use an open-path laser and measure the intensity of scattered light from transiting particles and relate that to particle size utilizing Mie-Lorenz theory and assuming something about the particles (typically liquid and spherical). The instruments output measurements as cumulative binned counts of droplet diameter. Some instruments including certain versions of the CDP, Fast CDP (FCDP; SPEC, Inc.), and FAST-FSSP are also capable of providing the sizes and interarrival





times of individual particles. Cloud particle sizes and counts are used to construct size distributions and calculate higher distribution moments.

Several sources may contribute to OPC sizing and counting errors that in turn propagate through to higher moments (Dye
and Baumgardner, 1984; Baumgardner et al., 1985; Cooper, 1988; Baumgardner and Spowart, 1990; Brenguier et al., 1998; McFarquhar et al., 2017; Wendisch and Brenguier, 2013). Sizing error can also result in artificial broadening of hydrometeor size distributions which can mistakenly be attributed to distribution-modifying cloud processes (Baumgardner et al., 1990). The non-linear relationship between droplet diameter and the intensity of light scattered by a droplet limits the resolution of size bins (Pinnick et al., 1981). The CDP has a default bin width of 2 μm for diameters larger than 14 μm, which can result in
as great as 15% uncertainty in diameter (Nagel et al., 2007). Mie resonance, which is more pronounced for the CDP's unimodal laser, also introduces sizing uncertainty for droplet diameters smaller than 14 μm (Knollenberg, 1976; Nagel et al., 2007).

Laser intensity and droplet scattering angles vary based on droplet transit location (Dye and Baumgardner, 1985; Brenguier
et al., 1998). Therefore, droplets are only counted and sized if they pass through the qualified sample area; an elliptical region within the depth of field where laser intensity and droplet scattering angles are relatively homogenous. Nonetheless, laser intensity and scattering angles are somewhat variable even within the qualified sample area, resulting in counting and sizing error that is dependent on droplet transit location (Brenguier et al., 1998; Wendisch et al., 1996). Instrument-specific misalignment of optical components can increase spatial variability in counting and sizing accuracy (Lance et al., 2010). The
cross-sectional area of the depth of field (DOF) is included in calculations of sample volume, such that errors in DOF will propagate as a scaling bias in concentration (Wendisch et al., 1996).

Coincidence error is a concentration-dependent phenomenon that occurs when multiple droplets are simultaneously within the sensitive area of an OPC's laser. Coincidence can affect sizing and counting accuracy but errors can be difficult to
characterize because they depend on many factors including particle concentration, particle size, the location that particles transit the laser, and instrument optical design (Baumgardner et al., 1985; Cooper, 1988; Brenguier, 1988).

FSSP electronic limitations require an 'electronic delay sequence' for a period after particle detection. Particles passing through the qualified sample area during the delay sequence are not detected resulting in undercounting, or 'dead time
losses', which requires algorithmic corrections to FSSP-measured concentration (Baumgardner et al., 1985; Brenguier et al., 1998; Baumgardner and Spowart, 1990; Brenguier et al., 1998). Newer OPCs including the FAST-FSSP, CDP, and FCDP feature faster electronics that negate dead time loss errors (Brenguier et al., 1998).



Forward scattering OPC measurements require that several assumptions be made about cloud particles. OPC techniques assume that measured particles are primarily composed of water and therefore have refractive indexes equal to that of pure water (Pinnick et al., 1981). Particles must also be small enough (less than ~50 μm diameter) to follow Mie-Lorenz scattering theory for the laser wavelength of an instrument. Particle shape affects scattering behaviour, so it is assumed that

liquid hydrometeors are spherical (Nagel et al., 2007). Several researchers have used the FSSP to study ice hydrometeors but such measurements are subject to uncertainty imposed by the variability in ice particle shape (Gardiner and Hallett, 1985; Field et al., 2003).

In mixed and ice phase cloud, ice particles are prone to shattering on contact with OPC structures. If passed through the

sample area, ice fragments can be erroneously identified as natural particles leading to errors in counting and sizing and an artificial bimodality in hydrometeor distributions (Gardiner and Hallett, 1985; Korolev and Isaac, 2005). FSSP measurements can be greatly affected by shattering artifacts because the probe's laser is housed in a cylindrical shroud (Heymsfield, 2007; McFarquhar et al., 2007). The CDP features an open-path laser that is passed between two arms that are often outfitted with anti shattering tips. As a result, particle shattering introduces negligible uncertainty in CDP

measurements, as demonstrated in work by Lance et al. (2010) and Khanal et al. (2018).

The FSSP and CDP are often calibrated by passing glass microbeads or polystyrene spheres through the sample area. These methods have crude control of calibration media placement and concentration such that they are only capable of testing OPC sizing response. Because these methods have limited control of particle concentration, coincidence can compromise

calibrations (Wendisch et al., 1996). Furthermore, the refractive indices for glass and polystyrene differ from that of water requiring a correction be applied to calibration measurements (Nagel et al., 2007). Wendisch et al. (1996), Korolev et al. (1991), Nagel et al. (2007), and Lance et al. (2010, 2012) developed droplet generating calibration systems that can produce and precisely place a mono-disperse stream of droplets of a known size/frequency at discrete locations within an instrument's sample area. These systems can test locationally-dependent sizing/counting accuracy at specific locations

throughout an instrument's sample area and in turn provide measurements of sample area dimensions.

This work uses a water droplet generating system to quantify CDP errors in counting and sizing resulting from variations in droplet scattering angles and laser intensity and misalignment of optical components. Seven droplet generator experiments with droplets of 9 – 46 μm in diameter provide data for detailed evaluations of CDP performance at locations throughout the

sampling area of the probe. This work is similar to the earlier work reported by Lance et al. (2010), but utilizes a wider range of droplet sizes over the entire qualified sample area and a much higher resolution of measurements across that area. Estimates of how errors in sizing and counting affect higher order moments are provided. Comparisons of in-situ CDP-derived liquid water content (LWC) and bulk LWC measurements from a hotwire device provide an additional means of evaluating probe performance.



## 2 CDP Operating Principles

The CDP features two forward-protruding arms; one houses a 568 nm laser diode and the other contains a series of collecting optics and photodetectors. As the probe is flown through cloud, some droplets transit the laser and scatter energy. The
collecting optics capture energy scattered by droplets in an ~12° arc, remove photons in the innermost ~4°, and focus the remaining energy onto a beam splitter. The beam splitter divides the laser energy and passes it to a sizer photodetector that is covered by an 800 μm diameter pinhole mask and a qualifier photodetector that is masked by a rectangular slit. Responses from the two photodetectors are converted to digital counts ranging from $1 - 4095$ counts. Sizer responses are used to estimate droplet diameter through Mie-Lorenz theory. The qualifier's rectangular mask is designed to reduce the collection
angles of the detector so that responses are maximized when droplets pass through the qualified sample area. A droplet is considered to be within the qualified sample area (or a "qualified droplet") if the signal from the qualifier is greater than one-half of the signal from the sizer. The CDP employs a dynamic sizer signal threshold in order to minimize false counting events resulting from impinging solar radiation or other sources of noise. This is accomplished by considering all sizer responses within a 10 Hz period that result in less than 512 digital counts. A noise band is defined as the region that contains
at least 75% of responses with less than 512 counts. Sizing/counting events are rejected if sizer response is less than the determined noise band (Lance et al., 2010; Droplet Measurement Technologies, 2014).

Standard coincidence occurs when multiple droplets are simultaneously within the qualified sample area of a CDP. OPCs are designed to count/measure a single particle at a time so standard coincidence results in undercounting and can also lead to
oversizing due to the additional light scattered by coincident hydrometeors (Baumgardner et al., 1985; Cooper, 1988). Because the qualified sample area of the CDP laser is relatively small (on the order of 0.3 mm$^2$), standard coincidence is only expected to affect measurements in regions of high droplet concentration (Lance et al., 2010).

Originally, the sizer detector was unmasked, meaning that it was sensitive to light scattered by droplets transiting through a
region surrounding the qualified sample area, called the extended sample area. Droplets passing through the extended sample area cause insignificant qualifier detector responses so they are not counted or sized. A specialized form of coincidence, called extended coincidence, occurs when droplets are simultaneously within the qualified and extended sample areas (Lance et al., 2010 and 2012). Coincident droplets within the extended sample area scatter additional light that can in turn result in oversizing of qualified droplets. Extended coincidence can also lead to undercounting if sizer response exceeds a threshold
value. Lance et al. (2010) used a droplet generating calibration system to measure the qualified and extended sample areas ($SA_Q$ and $SA_E$) using droplets with 12 and 22 μm diameters. The researchers found that $SA_E$ can be much larger than $SA_Q$ (20.1 mm$^2$ vs. 0.3 mm$^2$) resulting in errors from extended coincidence up to 60% oversizing and 50% undercounting in



concentrations as low as 400 cm$^{-3}$ (Lance et al., 2010). Results from Lance et al.'s 2010 study motivated the addition of an 800 μm diameter sizer pinhole mask that decreases the size of the extended sample area to ~2.7 mm$^2$, thus reducing the occurrence of extended coincidence (Lance et al., 2012). It was concluded that extended coincidence introduces negligible uncertainty in droplet concentrations less than 650 cm$^{-3}$ for CDPs featuring the sizer mask modification.

Lance et al.'s (2010) droplet generator work also tested CDP sizing and counting accuracy throughout the qualified sample area (at a spatial resolution of 200 x 20 μm) using 12 and 22 μm droplets. Ten additional tests investigated sizing accuracy at the centre of the qualified sample area using droplet diameters of 8 – 35 μm. It was shown that droplets are systematically oversized by 2 μm at the centre of the qualified sample area and that sizing accuracy for 12 and 22 μm droplets is dependent

upon where droplets transit the qualified sample area. Droplets were undersized by as much as 74% in certain sample locations and oversized by as much as 12% in others (Lance et al., 2010). It was found that on average, 12 and 22 μm droplets were counted to within 95% accuracy. Counting error is more severe at the edges of the qualified sample area as a result of photodetector signal noise (Lance et al., 2010).

**3 University of Wyoming droplet generating system**

The University of Wyoming (UW) Atmospheric Science Department developed a droplet generating calibration system very similar to the system built by Lance et al. (2010, 2012) which is based on work by Korolev et al. (1991), Wendisch et al. (1996), and Nagel et al. (2007). A detailed explanation of the design and operation of a droplet generating system can be found in Lance et al. (2010). The systems built by the UW team and Lance et al. (2010, 2012) use a piezoelectric print head to produce a mono-disperse stream of water droplets inside a glass flow tube. The flow tube contains a sheath flow that is

accelerated in a tapered exit region, which by extension, accelerates and focuses suspended droplets into a precise stream. The accelerated droplet stream is then passed through a CDP's sample area at discrete locations. Two-axis positioning stages are used to control the point of droplet injection and provide the coordinates of injection locations. The UW droplet generator incorporates computerized stages, instead of the manual positioners used by Lance et al. (2010), in order to automate and expedite the testing procedure.

A high speed metrology camera outfitted with a 10X microscope objective provides an independent measurement of droplet diameter using the glare technique as described by Korolev et al. (1991), Wendisch et al. (1996), Nagel et al. (2007), and Lance et al. (2010). As droplets pass through the laser of the CDP, the left and right sides of the droplet are illuminated as a result of reflection and refraction. The metrology camera images these illuminated regions (glares) which appear as two

parallel lines when using an exposure time of 1/1000 sec. Estimates of droplet diameter are obtained by considering the pixel separation of glares, a pixel to distance conversion, and a formula that accounts for the angle of the camera objective relative



to the laser (Wendisch et al. (1996) and Korolev et al. (1991). Using this technique, the UW system is capable of determining individual droplet diameters to within ±0.355 μm.

The metrology camera can also be used to estimate droplet velocity by capturing images with exposures on the order of

1/150,000 – 1/300,000 second. Shorter exposure times produce glare images with well-defined start and end points. Droplet velocity can be estimated by considering glare length, a pixel to distance conversion, and exposure time. The longitudinal position of glares can also be used to evaluate droplet placement precision.

A number of validation tests were performed to ensure that the UW droplet generator can produce droplets of consistent

diameter for the amount of time required to conduct a test (~4 hours), precisely place droplets at discreet locations within the sample area, and eject droplets at suitable velocities. Seven droplet generator tests that produced droplet diameters of 9 – 46 μm are used to evaluate accuracy and consistency in droplet diameter. During the course of each test, glare images were captured once every second and a random sample of 80 images were analysed to provide distributions of true droplet diameter ($D_{true}$). Table 1 shows that standard deviation of $D_{true}$ is less than 0.7 μm for all seven tests. It also shows that all but

one test produce $95^{th} – 5^{th}$ percentile range of $D_{true}$ less than the 2 μm bin width of the CDP (for droplets larger than 14 μm).

Two tests were conducted that used the deviation of glare position to validate droplet placement precision. To confirm that placement precision is similar along orthogonal axes, glare images of 32 μm droplets were captured for 1 hour with the metrology camera placed at 124.9º incident to the CDP laser and an additional hour at 214.9º incident. Glare position for a

random sample of 50 images from each camera angle show that droplet deviation is similar along orthogonal axes. The absolute deviation of glares is 5.7 μm along both axes and standard deviations are 1.5 and 1.7 μm for the 124.9º and 214.9º camera angles respectively. A separate experiment tested long term placement precision by analysing 80 random glare images captured over the course of a four-hour test. Droplet position for the sample has an absolute range of 11.4 μm and a $95^{th} – 5^{th}$ percentile range of 9.3 μm. Approximately 8% of droplets were placed beyond 10 μm.

Droplet ejection velocity is validated by capturing images using exposure times of 1/150,000 – 1/300,000 sec. It was found that when droplets were created and accelerated in a 13 l min$^{-1}$ sheath flow, 40 μm diameter droplets cross the CDP laser at ~32 m s$^{-1}$. This velocity is only about 30% of typical University of Wyoming King Air research airspeeds but is greater than the minimum operational airspeed of the CDP (10 m s$^{-1}$).



## 4 Results of Droplet Generator Tests on the CDP

### 4.1 Experimental design

To quantify uncertainty in CDP measurements of droplet counting and sizing, seven tests using nominal droplet diameters of 9, 17, 24, 29, 34, 38, and 46 μm provided measurements over most of the size range detectable by the CDP. For each test,
droplets were injected at fixed locations through the qualified sample area of the CDP. Droplets were injected at a frequency of 200 Hz for 9 μm droplets and 250 Hz for all other sizes. Following a dwell time at a given location, the position of the droplet injector relative to the CDP sample area was moved a small distance. The tests proceeded in this fashion, injecting droplets throughout the entire qualified sample area of the CDP. The start and end times at each location were recorded. Post-test, 5 Hz data from the CDP were synchronized to match droplet location and CDP measurements.

The time required to complete a full test was in some cases as long as 5 hours (see Table 1). Stability of the droplet generator system over this time depends, in part, on the size of droplets being produced. Smaller droplets tend to result in reduced system stability and therefore required shorter test periods. For the five tests using droplets 24 μm and larger, the dwell time at each sample location was 2 seconds. This resulted in 500 droplets passing through the CDP sample area at each location.
For these same tests, a 10 μm by 10 μm grid of sample locations covered the entire test area, corresponding to 2700 discrete sample locations across the approximately 0.27 mm$^2$ qualified sample area of the CDP. For the test using 17 μm droplets, the dwell time at each location was reduced by a factor of 2 and the grid resolution remained the same. The system was less stable when producing 9 μm droplets and required test times of less than 2 hours to ensure consistent droplet sizes and placement throughout the experiment. For this test, dwell time was further reduced such that 200 drops were placed at each
location and the resolution of the grid was reduced to 30 μm by 20 μm, resulting in roughly 450 discrete locations across the qualified CDP sample area.

### 4.2 CDP Sizing

CDP measurements for all droplets detected during a given test were used to produce a distribution of droplet diameters for that test. Droplet distributions were computed using number counts from each of the CDP's 30 pre-determined size bins. Bin
widths are 1 μm for diameters less than 14 μm, and 2 μm for diameters greater than 14 μm. For each bin, we considered the geometric mean diameter, hereafter referred to as $D_{CDP}$. Also, for each test, 80 randomly selected droplet glares were analysed to determine a distribution of actual droplet diameters, $D_{true}$. These droplets, when binned according to CDP size bins, resulted in a distribution of droplets, $D_{true}$*.

Figure 1 shows distributions of normalized frequency for $D_{CDP}$ and $D_{true}$* for each of the seven tests. In general, the mode diameter of the distribution based on sizing from the CDP ($D_{CDP}$) was within one to two size bins (1 to 4 μm) of the $D_{true}$* mode. For the test using 9 μm droplets, more than 50% of the droplets detected by the CDP were placed in the 7.5 μm bin



and another 30% were placed in the 8.5 μm bin. Nearly 90% of the randomly selected droplets were determined to have actual diameters between 8 and 10 μm, suggesting that the CDP undersized droplets in this range by about 1 to 2 μm. Table 2 shows that the absolute difference between mean $D_{CDP}$ and mean $D_{true}$* was 1.3 μm.

Tests using 17 and 24 μm diameter droplets resulted in a better match between $D_{CDP}$ and $D_{true}$*. For each test, the medians and modes of $D_{CDP}$ and $D_{true}$* were in the same bin and more than 95% of the droplets were contained in the same two bins. However, the breadth of the distribution measured by the CDP was slightly larger than the actual distribution for the 17 μm test. And, for the 24 μm test, the distribution measured by the CDP was skewed to smaller sizes. For both tests, absolute differences between the means of the distributions were less than 1 μm (Table 2).

For the 29, 34, 38, and 46 μm diameter tests, a steady trend of oversizing with increasing droplet diameter is apparent when comparing the normalized histograms of $D_{CDP}$ and $D_{true}$* (Fig 1). In all cases, the mode diameter from CDP measurements was one bin larger than the true diameter mode. For the largest droplet test, 46 μm, 55% of the droplet diameters from the CDP fell in the 48 to 50 μm bin and another 10% fell in each of the 44 to 46 and 46 to 48 μm bins. More than 95% of the

actual diameters were split roughly equally between the 44 to 46 and 46 to 48 μm bins.

Skewing of the CDP-measured distribution to smaller sizes occurred for all tests using droplets 24 μm and larger. Further, the breadth of the CDP-measured distribution increased with increasing droplet diameter. This is perhaps more apparent from the data in the last column in Table 2. Here we compare the difference between the 95[th] and 5[th] percentile range for

$D_{true}$* and $D_{CDP}$. The difference increased significantly for larger diameter tests. Interestingly, even though the difference in both the mode and median diameters of the CDP-measured distributions (compared to $D_{true}$*) were larger for these larger diameters, the absolute difference between the means of the distributions were quite small, roughly 0.1 to 0.2 μm. This is because, with the measured distributions skewed to smaller diameters, comparisons of mean diameters appeared to compare more favourably.

By matching the measured response of the CDP to the expected Mie scattering curve, it is possible to investigate whether the errors in sizing observed from the droplet generator tests may be accounted for by limitations due to Mie resonances or by uncertainty in scattering angle collection. The CDP's nominal collection angles are 4° to 12°. However, optical misalignment along with the physical dimensions of the probe's depth of field may lead to uncertainties up to 1° (Baumgardner et al.,

2017). Figure 2 shows that the Mie response curve matches reasonably well with the CDP threshold counts that are used to sort droplets into discrete size bins. Two ranges of scattering angles are considered and both show similar behaviour. In fact, regardless of which range of angles are considered, the error in sizing is expected to be, on average, nearly the same. Errors in drop sizing for individual drops, however, will vary depending on collection angles.





The shaded region in Figure 2 illustrates the range of threshold counts that the CDP uses to determine the size bin for an individual drop. Regions where the Mie curve(s) lie within the shaded regions are locations where a drop will be sized 'correctly'. If the Mie curve is above the shaded region, the drop will be oversized; below the shaded region it will be undersized. The amplitude of the Mie resonances and the locations of the peaks and valleys depend on droplet diameter and

vary with collection angles. Generally, the amplitude of the Mie resonances increases with increasing drop size, however, so does the 'steepness' of the curve. Therefore larger droplets, 40 to 50 μm in diameter, should not be undersized or oversized by more than about 2 μm. However, smaller droplets less than about 20 μm in diameter may easily be mis-sized by more than 2 μm, accounting for as much as ±20% error in sizing (Baumgardner et al., 2017).

Results from the droplet generator tests overlaid on Figure 2 provide additional insight into CDP response. The mean and 5[th] to 95[th] percentile range of $D_{true}$ illustrates that the droplets being produced nearly all fell within one size bin of the CDP for any given test. The corresponding MIE resonance curves over those same size ranges generally fluctuate over a range of A/D counts that correspond to threshold values of up to 2 to 3 size bins. This can be seen by examining the Mie response (4° - 12°) for the test producing 29 μm drops. Over the range of droplet sizes produced, some locations of the Mie curve fall just

below the threshold box (for 28-30 μm bin), while others fall slightly above and still other fall inside the box.

The skewing of CDP-measured distributions to smaller sizes is also apparent by examining the CDP response compared to the threshold curves. For each test, the mean value of A/D counts (Figure 2) lies either within or very near the appropriate threshold box for that droplet diameter. However, the median value of A/D counts exceeds the threshold box for that droplet

diameter for all tests using droplet diameters 29 μm and greater. This suggests that the calibration of the CDP is based upon *mean* diameter of drops rather than the median or mode diameter. While this may be appropriate, because of the unnatural skewing to smaller sizes, it does have implications on calculations of higher order moments. The severe undersizing of a small sample of drops for these same tests cannot be explained based on Mie resonance or collection angle considerations.

Figure 3 illustrates sizing from the CDP across the entire qualified sample area. For each sample location, the mean of $D_{CDP}$ is compared to the mean of $D_{true}$*. A positive difference (warm colours) indicates oversizing by the CDP, negative values (cool colours) indicate undersizing. For the 9 μm diameter tests, undersizing of droplets by 1 to 2 μm was found throughout most of the CDP sample area. Droplets passing through the centre of the beam and laterally towards the top experienced somewhat less undersizing than in other regions. No regions indicated oversizing of droplets. For the 17 μm tests, droplets

throughout much of the sample area were sized correctly. Only a small region, laterally towards the top of the beam and towards the detector, were droplets oversized, on average by about 1 μm.

The five remaining droplet tests, 24 μm and larger, all revealed a similar behaviour. In all cases, there was a lateral dependence on sizing from the top to the bottom of the beam along the entire length of the qualified sample area. The




magnitude of the sizing difference laterally across the beam increased with increasing droplet size. For the 24 μm test, the sizing difference was only about 2 μm across the beam, but for the 46 μm test, the sizing difference was nearly 6 μm across the beam. Also, for each of these five tests, a region near the detector showed significant undersizing of droplets that also increased in magnitude with increasing droplet size. For the 46 μm test, droplets were undersized by as much as 30 μm. This

region accounts for the skewing to smaller sizes of the distributions discussed earlier in this section.

Columns three and four in Table 2 provide information about how sizing differences for each test impact higher moments of the droplet size distribution. For the 9 μm test, the volume-weighted mean diameter (VMD) measured by the CDP was 1.1 μm small compared to that computed from the $D_{true}$* distribution, resulting in a 36.7% underestimate in LWC. For the 17 and

24 μm tests, the absolute difference between the actual and measured VMD was less than 0.25 μm and resulted in a roughly 8% overestimate and 2% underestimate in LWC for these droplets, respectively. For tests using droplets larger than 24 μm, the CDP oversized VMD from 1 to 1.5 μm, resulting in overestimates of LWC of 2.4 to 11%. Readers should note that errors in sizing by a given amount will have a much more significant impact on LWC for smaller droplets. However, for real measurements in cloud, it is often the larger droplets that carry the majority of the liquid mass. Therefore, these middle and

larger sizes from 20 μm and greater are expected to have the greatest impact on LWC estimates from the CDP.

### 4.3 Counting accuracy and qualified sample area measurements

Counting accuracy is evaluated by comparing CDP-recorded counts to the actual number of droplets based on print head ejection frequency and dwell time at each sample location. For all tests, droplets are counted to within 98% accuracy in ~95% of the sample locations. Experiments indicate that all sizes of droplets are undercounted around the perimeter of the

qualified sample area, presumably as a result of sizer and qualifier signal noise (Lance et al., 2010). Figure 4 shows locationally-dependent counting accuracy for 46 μm droplets where purple areas correspond to locations where the CDP recorded 10 - 50% of actual counts, blue show locations where 50 – 90% of actual counts were recorded, and green denotes where at least 90% actual counts were recorded. Only 46 μm droplets were overcounted, specifically in two isolated regions where droplets were overcounted by as much as 100%. The regions are located just left of the area where 46 μm droplets

were significantly undersized (see Fig. 3g and discussion earlier). Overcounting in these regions contributes to less than 1% overall count error because they occupy less than 1% of total SA_Q.

Figure 5 shows SA_Q calculated by summing the individual areas of sample locations that received a certain percentage of actual counts. SA_Q is calculated three times for each test by constraining which sample locations are considered to those that

received at least 10, 50, and 90% actual counts (SA_{Q_10%}, SA_{Q_50%}, SA_{Q_90%}). Evaluating SA_Q using this count threshold method provides uncertainty ranges of SA_Q and accounts for the fact that ~8% of droplets were placed beyond sample area bounds.




The mean value of $SA_{Q\_50\%}$ considering all tests is 0.269 mm$^2$; compared to a value of 0.30 mm$^2$ provided by the manufacturer. $SA_{Q\_50\%}$ varies 0.03 mm$^2$ across the range of droplet diameters tested. It is smallest for 9 and 17 μm droplets, reaches a maximum of 0.28 mm$^2$ for 24 μm droplets, and then decreases to 0.27 mm$^2$ for 46 μm droplets. The range of $SA_{Q\_10\%}$ to $SA_{Q\_90\%}$ is smallest for the largest droplets, most likely because detector noise is less of a consideration for larger

droplets that scatter relatively more light and hence provide a greater detector response. The test using 9 μm droplets shows the greatest difference between $SA_{Q\_10\%}$ and $SA_{Q\_90\%}$, but it should be noted that $SA_Q$ variability is likely exaggerated by the course spatial resolution used for that experiment.

For calculations of number concentration and higher moments, $SA_Q$ can be provided by either using a fixed value equal to
the mean for all droplet sizes (solid red line in Figure 5) or by using a variable value based on a second degree polynomial fit (blue curve in Figure 5). To explore the impact of employing a fixed vs. variable $SA_Q$, three Poissonian droplet distributions with means of 10, 25, and 35 μm are prescribed. The concentration of each distribution equals 100 cm$^{-3}$ when calculated with a fixed $SA_{Q\_50\%}$ of 0.27 mm$^2$. Table 3 illustrates how using a fixed vs. variable $SA_Q$ affects concentration and LWC. It shows that choice of $SA_Q$ type most affects concentration and LWC for the distribution with a 10 μm mean diameter. Using a
variable $SA_Q$ results in 6% greater concentration and ~4% greater LWC. For distributions with greater mean diameters, the choice of using a fixed or variable $SA_Q$ results in less than 3% difference in concentration and LWC.

It seems best to calculate higher moments using a fixed $SA_Q$ of 0.27 mm$^2$, given that the choice of $SA_Q$ type has relatively little impact on concentration or LWC. Furthermore, the second-degree polynomial fit used to model variable $SA_Q$ does not
completely capture variations in $SA_Q$ for droplets with diameters between 20 to 30 μm and requires extrapolation of $SA_Q$ for droplets with mean diameter less than 9 μm or greater than 46 μm.

## 5 Comparisons of liquid water content from the CDP and Nevzorov probes

In-situ data collected by aircraft are used to further investigate uncertainty in CDP measurements by comparing LWC derived from CDP measurements to bulk LWC measured by a Nevzorov hotwire probe. Comparisons of in-situ LWC
measurements provide an independent evaluation of CDP performance and an indication of how error in real-world CDP measurements compares to laboratory droplet generator results.

### 5.1 The University of Wyoming King Air

The University of Wyoming King Air (UWKA) is a Beechcraft Super King Air modified to carry a variety of atmospheric in-situ and remote sensors capable of collecting information about atmospheric thermodynamics, dynamics, and cloud
particle properties (Wang et al., 2012). In the following, we utilize measurements from two field campaigns conducted in late 2016 and early 2017. The Precipitation and Cloud Measurements for Instrument Characterization and Evaluation



(PACMICE) campaign began in August 2016 and lasted until May 2017, with flights over eastern Wyoming and western Nebraska, USA. It focused on collecting cloud and precipitation measurements in precipitating stratiform and convective systems primarily in the shoulder seasons. The Seeded and Natural Orographic Wintertime clouds – the Idaho Experiment (SNOWIE) occurred during January – March 2017, and focused on wintertime orographic clouds in southwestern Idaho,

USA (French et al., 2018). The majority of clouds sampled in both PACMICE and SNOWIE were mixed phase.

**5.2 Constant temperature hotwire probes**

The UWKA carries both a DMT LWC-100 and a deep-cone Nevzorov constant temperature hotwire probe. Both provide estimates of bulk cloud water content utilizing changes in current supplied to heated elements that are exposed to impacts of cloud particles (King et al., 1978; Baumgardner et al., 2017). Element temperature is maintained near 100 C such that

impinging particles will vaporize transferring energy from the element through the effects of sensible and latent heating. Control circuitry maintains element temperature by altering the power supplied using element resistance as a proxy for temperature. Measurements of water content are obtained by relating the power required to maintain element temperature as particles are vaporized to the sensible and latent heat capacities of water, and element surface area (King et al., 1978; Korolev et al., 1998).

Convective losses due to moist airflow over the sensor also transfer energy from collector elements and can be quite large at aircraft flight speeds (King et al., 1978; McFarquhar et. al, 2017). The Nevzorov probe features reference elements that are positioned on the devices' trailing edge such that they are aerodynamically shielded from particle impact (Korolev et al., 1998; Strapp et al., 2003). Energy losses from the reference elements are then assumed to arise solely due to convective

considerations and thus the total power delivered to the reference elements can be used to estimate the convective heat losses from the sensing (collector) elements. The relationship between collector and reference element convective losses depends on airspeed and density (Korolev et al., 1998, Abel et al., 2014). Data collected during clear air calibration manoeuvres are used to compute the ratio of collector to reference power and determine how the ratio varies with airspeed and density. The manoeuvres are typically flown at several flight levels over a range of airspeeds. Any inaccuracy in the estimate of

convective heat losses in the collector sensor based on power delivered to the reference sensor results in baseline drift of the Nevzorov-derived LWC ($LWC_{NEV}$) measurement (Abel et al., 2014). For the data used herein, the effectiveness of the Nevzorov data processing method was evaluated using ~60,000 out-of-cloud points. $LWC_{NEV}$ residual (i.e. departure from zero when not in-cloud) was used to determine uncertainty in baseline LWC. $LWC_{NEV}$ baseline uncertainty is estimated to be no greater than 0.05 g m$^{-3}$ (the 95$^{th}$ – 5$^{th}$ percentile range of residual LWC) and minimum detectable $LWC_{NEV}$ is +0.02 g m$^{-3}$

(95$^{th}$ percentile residual LWC).

The Nevzorov is capable of measuring both LWC and total condensed water content (TWC) using two collector elements with different geometrical designs (Korolev et al., 1998). Estimates of ice water content (IWC) can then be obtained by





differencing the two measurements. The LWC element is in the shape of a thin rod designed to only evaporate liquid particles. Ice particles shatter on impact with the sensor and are swept away before significant melting or evaporation can occur (Korolev et al., 1998). The TWC collector has a 'deep inverted cone' shape designed to capture both liquid and ice particles (Korolev et al, 2013). Korolev et al. (1998) showed that in mixed phase conditions, interactions between the LWC

collector and ice particles can result in LWC overestimation on the order of 12% IWC.

In some conditions, collection efficiency may be significantly less than unity resulting in underestimation of $LWC_{NEV}$. Because airflow diverges in the vicinity of the LWC collector, $LWC_{NEV}$ may be underestimated by as much as 30% in droplet populations with VMD less than 8 μm since particles with insignificant mass are unable to cross the divergent

streamlines and impact collector elements (Korolev et al., 1998). Collection efficiency also departs from unity for droplets with VMD greater than 30 μm because larger droplets tend to splatter on impact leading to incomplete evaporation (Schwarzenboeck et al., 2009).

**5.3 Dataset overview**

Data were used from 29 research flights from both PACMICE and SNOWIE. Measurements from both probes were filtered

to 1 Hz. Here we select only those data points in which both $LWC_{NEV}$ and $LWC_{CDP}$ exceeded a threshold value of 0.05 g m$^{-3}$. To minimize uncertainty due to presence of ice hydrometeors in CDP and Nevzorov LWC measurements, the IWC from the Nevzorov was used to select periods of liquid phase only penetrations from PACMICE and SNOWIE missions. However, IWC estimates from the Nevzorov may be affected by as-of-yet uncharacterized sources of uncertainty such that one cannot conclude the dataset used here is completely devoid of mixed-phase penetrations. Nonetheless, uncertainty in LWC resulting

from the presence of ice is expected to minimally impact results. $LWC_{NEV}$ is subject to overestimation of less than 12% IWC, which is often small compared to LWC in mixed phase cloud. It has been also been demonstrated that the CDP is minimally effected by ice shattering artifacts (Lance et al., 2010, Khanal et al., 2018).

The resultant data subset used in the comparison contains 17,917 1 Hz in-cloud points. Droplet concentrations encountered

during SNOWIE were uncharacteristically low for continental clouds. Mean droplet concentration for the dataset is 113.6 cm$^{-3}$ with 50% of data points having concentration less than 50 cm$^{-3}$. Consequently, droplets were relatively large; with an average VMD of 22.2 μm and 1$^{st}$ and 3$^{rd}$ quartiles of 16.7 and 27.7 μm. Nearly all measurements were taken in supercooled conditions, the environmental temperature range for the 5$^{th}$ and 95$^{th}$ percentile is -18.7 and -1.3 °C.

**5.4 In-situ results**

For each 1 Hz data point, measured spectra from the CDP were used to compute the total droplet concentration and the VMD of the spectra. The data were first subdivided based on droplet concentration and then further divided based on VMD. Figure 6 shows percent difference between $LWC_{CDP}$ and $LWC_{NEV}$ as a function of binned values of VMD (positive values indicate



LWC$_{CDP}$ is greater than LWC$_{NEV}$) for four different ranges of droplet concentration. The points represent mean percent LWC difference calculated using a best fit line for all points in 5 μm wide bins (5-10 μm, 10-15 μm, 15-20 μm, etc.); error bars are root mean squared error. Green hatched areas are estimates of expected percent LWC difference using droplet generator results computed for errors in CDP sizing and counting for 10% and 90% count thresholds and errors considering best

estimates of Nevzorov collection efficiency as a function of VMD (from Korolev et al., 1998; Strapp et al., 2003; Schwarzenboeck et al., 2009). Plots are shown for 4 ranges of total droplet concentration. The mean percent LWC difference and the number of data points (n) are shown in the bottom-right corner of each plot.

For all VMDs larger than 10 μm, LWC$_{CDP}$ exceeded LWC$_{NEV}$ by as much as 40%. For VMDs less than 10 μm, LWC$_{CDP}$ was

less than LWC$_{NEV}$ by 5 – 10% for those cases in which total droplet concentrations were less than about 400 cm$^{-3}$. The general trend, for all droplet concentrations, suggests increasing LWC$_{CDP}$ (compared to LWC$_{NEV}$) for increasing VMD. However, the mean difference in LWC, across all VMDs, does not indicate any specific trend when considering different ranges of total droplet concentration.

Estimates of percent LWC difference expected based on results from droplet generator tests and Nevzorov collection efficiency estimates predict that LWC$_{CDP}$ should be at most 11% greater than LWC$_{NEV}$. However, when all of the data in this study are considered, the mean percent difference is 19.6%. Two striking features of the data show that the percent difference for large VMD, greater than about 25 – 30 μm, is considerably larger than expected. And, for droplet concentrations greater than 400 cm$^{-3}$, the percent difference is significantly larger than expected for all VMDs. The larger

than predicted difference between LWC$_{CDP}$ and LWC$_{NEV}$ is unlikely to be a result of coincidence error. The UWKA CDP features a sizer pinhole mask modification such that it is expected to be relatively unaffected by coincidence in concentrations less than 600 cm$^{-3}$ (Lance et al., 2012). Figure 6d shows that mean percent LWC difference for data with concentration of 400 – 1600 cm$^{-3}$ is not significantly different than mean values for much smaller total droplet concentrations. On the other hand, for this concentration range, percent LWC difference is significantly larger for smaller

VMDs when compared to similar VMDs for lesser droplet concentrations, suggesting that those CDP measurements may indeed be impacted by coincidence for these higher concentrations. Regardless, these data account for less than 4% of all points and suggest coincidence is unlikely to account for differences across all ranges of concentration.

The droplet generator tests used a near constant droplet velocity that was ~30% of typical UWKA airspeeds.  They provide

no information about how CDP sizing/counting accuracy and SA$_Q$ may vary with airspeed. Some of the discrepancy between estimated and actual percent LWC difference could be a result of a change in CDP performance at typical aircraft flight speeds which could result from limitations in photodetector response (Dye and Baumgardner, 1984). However, in order for airspeed-dependent errors in sizing and/or SA$_Q$ to account for the discrepancies shown, increased flight speeds would need to result in an increase in sizing (and hence photodetector output for the sizer signal) and/or an increase in SA$_Q$, both of which





are unlikely outcomes; one might expect the opposite behaviour. On the other hand, overcounting could increase with increasing particle velocity if photodetector response limitations result in more significant signal noise. But it seems unlikely that such considerations could cause overcounting on the order of 5 - 20% given that only 46 μm droplets were overcounted (by less than 1%) during droplet generator tests.

It is possible that the discrepancy between estimated and actual percent LWC difference could be a result of a change in counting/sizing behaviour for droplets passing through the qualified sample area region where droplets are severely undersized (blue areas in the rightmost 10% of the beam maps shown in Fig. 3). Sizer responses are characteristically within the noise band range (less than 512 digital counts) for droplets transiting these regions; thus, severely undersized droplets

could be rejected during 'real-world' operation. If $LWC_{CDP}$ error estimates (as described in section 3.2) are recalculated excluding these regions where droplets are severely undersized, then the resultant oversizing throughout the rest of the sensitive sample area could result in as much as 17% overestimation of $LWC_{CDP}$ (effectively shifting upward the hatched green areas in Figure 6 for large VMD).

Error in Nevzorov measurements could also contribute to the discrepancy between $LWC_{CDP}$ and $LWC_{NEV}$. Instrument icing was a common issue during SNOWIE. The 0.05 g m$^{-3}$ threshold applied to $LWC_{CDP}$ and $LWC_{NEV}$ was used to exclude measurements taken when one (or both) of the instruments was (were) completely unresponsive. In the case of ice accumulation on the Nevzorov sensing element, build-up of rime ice near (or over) the LWC element often results in significant baseline drift along with an accompanying reduction in sensitivity to liquid water (due to changes in airflow and

shielding of the sensing element). Such situations would result in an underestimation of LWC by the Nevzorov and could explain some of the differences shown in Figure 6. However, examination of baselines prior to and after exiting clouds suggests this is not a large problem for the cases examined. Regardless, nearly all measurements were obtained in supercooled conditions so the data used in this study were not able to be further subdivided to investigate differences in regions where temperature greater than 0 °C would exclude possibility of icing.

**6 Summary**

A droplet generating calibration system was used to test the sizing and counting performance and provide measurements of the qualified sample area of the UWKA CDP using seven discrete droplet diameters ranging from 9 – 46 μm. Experiments reveal that droplet sizing accuracy varies depending on where droplets transit the sample area and the size of the droplets. Errors in sizing for the majority of droplets across the size ranges tested can be accounted for by the amplitude of Mie

resonances on the response curve. The Mie resonances often result in an artificial broadening of the distribution by 1 – 2 bins. How much broadening occurs depends on droplet size and the actual range of collection angles for the probe. This finding confirms results of earlier studies (Rosenberg et al., 2012; Baumgardner et al., 2017). Droplets with nominal



diameter of 9 µm are undersized by 1 µm or less for roughly 33% of the droplets sampled and are undersized by 1 – 4 µm for the remaining 66% of droplets. Errors in droplet sizing for 9 µm droplets do not depend strongly on where droplets transited the sample area. The errors in sizing for these smallest droplets are likely related to the amplitude of Mie resonances compared to the relatively shallow slope of the Mie function. Droplets with diameters of 17 and 24 µm are sized to within 2

µm of the true droplet diameter for nearly all droplets sampled (>90%), but there appears to be a small lateral dependence within the sample area on errors in sizing, such that droplets passing through the top half of the sample area are sized larger than those transiting through the bottom half. Tests for droplets with diameters 29, 34, 38, and 46 µm reveal more significant oversizing, by as much as 2 – 4 µm, with an even stronger lateral dependence on sizing error. Droplet generator experiments performed by Lance et al. (2010) using 12 and 22 µm droplets reveal a similar lateral gradient in sizing accuracy. The

researchers attributed this behaviour to a misalignment of the qualifier detector mask, however this consistent behaviour across different probes might indicate a problem with the optical design. Similar tests on two other CDPs using the University of Wyoming Droplet Generator system revealed similar lateral dependencies.

The tests also reveal that for droplets 24 µm and larger, nearly all droplets passing through 10% of the qualified sample area

(that portion closest to the detector) are undersized, by as much as 30 µm, depending on the droplet diameter. However, droplets passing through much of the rest of the sample area are oversized. The locationally-dependent nature of sizing accuracy results in artificial spectral broadening of droplet size distributions, which is most pronounced for droplets with diameters 34 µm and larger. Although droplets are oversized by 2 – 4 µm in most locations within the qualified sample area, the resulting errors in higher order moments such as mean diameter, VMD, and LWC, are mostly offset by undersizing of

droplets throughout the rest of the sample area. This has implications for how sizing should be calibrated for the CDP. For example, matching distribution modes when performing calibrations will result in an underestimation for higher order moments because distributions are artificially skewed. Conversely, calibrations that match mean droplet diameter will result in an over estimation of the diameter of the droplet distribution mode in real clouds.

Droplets were counted to within 98% accuracy over roughly 95% of sample locations. Only the largest droplets tested, 46 µm, indicated any significant over counting. This occurred in two regions bordering the area where these same droplets were significantly undersized. However, these regions account for less than 1% of the total qualified sample area so they introduce less than 1% overall count error. All sizes of droplets are undercounted around the perimeter of the qualified sample area and this must be considered when defining $SA_Q$ for higher moment calculations. $SA_{Q\_50\%}$ varies only 0.03 mm$^2$ depending on

droplet diameter and thus the use of a mean of $SA_{Q\_50\%}$ for all droplet sizes (0.27 mm$^2$) is warranted. $SA_Q$ for the CDP used in this work and the CDPs examined by Lance et al. (2010, 2012) agree to within 10%.

Comparisons of in-situ LWC measurements from the CDP and a Nevzorov hotwire probe provide another means of evaluating CDP performance. In-situ comparisons show that, on average, $LWC_{CDP}$ is greater by about 20% whereas droplet





generator results and Nevzorov collection efficiency considerations predict that $LWC_{CDP}$ should be no more than 12% larger. Droplet generator tests used a droplet velocity that is ~30% of typical UWKA airspeeds. The discrepancy between the expected and actual $LWC_{CDP}$ and $LWC_{NEV}$ difference may be a result of CDP performance at higher droplet transiting speeds. It also cannot be ruled out that the discrepancy may be partially due to rime ice build-up on the Nevzorov element.

Further studies should include a concerted effort to increase droplet speed within the droplet generator and in situ measurements in cloud at temperatures greater than 0 °C.

**Author Contributions**

SF completed the bulk of this work and wrote the manuscript. This manuscript was based on SF's Master's Thesis from University of Wyoming. The work was supervised by JF. Significant edits to the original manuscript were completed by JF.

RJ constructed the early version of the droplet generator and worked closely on initial testing.

**Acknowledgements**

The author's wish to thank the crew of the University of Wyoming King Air for their dedicated work during the PACMICE and SNOWIE campaigns. They also wish to thank Mr. Ben Heesen at University of Wyoming for his aid in constructing the droplet generator system. Development of the droplet generator was paid for by internal funds from the University. UWKA

flights for SNOWIE were supported through National Science Foundation grant AGS-1441831.

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

**Tables**

Table 1. Droplet generator test characteristics including the number of droplets injected at each sample location, longitudinal and latitudinal resolution, test duration, mean droplet diameter from glares (mean $D_{true}$), and the 95th to 5th percentile range of $D_{true}$. $D_{true}$ statistics are from 80 randomly-selected glare images.

| Test | Droplets per sample location | Long. Res. [μm] | Lat. Res. [μm] | Duration [hour:min] | Mean $D_{true}$ [μm] | $D_{true}$ 95th – 5th percentile range [μm] |
|---|---|---|---|---|---|---|
| 9 μm | 200 | 30 | 20 | 01:43 | 9.0 ± 0.5 | 1.6 |
| 17 μm | 250 | 10 | 10 | 03:14 | 17.3 ± 0.5 | 1.6 |
| 24 μm | 500 | 10 | 10 | 04:37 | 24.4 ± 0.4 | 0.8 |
| 29 μm | 500 | 10 | 10 | 04:15 | 28.8 ± 0.7 | 2.3 |
| 34 μm | 500 | 10 | 10 | 03:57 | 33.6 ± 0.4 | 0.8 |
| 38 μm | 500 | 10 | 10 | 03:35 | 38.5 ± 0.4 | 0.8 |
| 46 μm | 500 | 10 | 10 | 04:44 | 46.0 ± 0.4 | 0.8 |



**Table 2. Comparisons of the difference in several distribution parameters when calculated using CDP-recorded droplet diameter ($D_{CDP}$) vs. diameter from glares rounded to the geometric mean of CDP size bins ($D_{true}$*). A positive difference (or positive percent difference) indicates that calculations using $D_{CDP}$ result in a larger value than $D_{true}$*. Percent LWC difference is calculated by comparing the integrated $3^{rd}$ moment of normalized $D_{CDP}$ distributions vs. normalized $D_{true}$* distributions.**

| Test | Mean $D_{CDP}$ minus Mean $D_{true}$* [µm] | Difference in VMD [µm] | % LWC difference [µm] | Difference in $95^{th}$ – $5^{th}$ percentile range [µm] |
|---|---|---|---|---|
| 9 µm | -1.3 | -1.1 | -36.7 | 0.0 |
| 17 µm | 0.4 | 0.2 | 8.2 | 2.0 |
| 24 µm | -0.3 | -0.1 | -2.0 | 2.0 |
| 29 µm | 0.8 | 1.3 | 11.1 | 4.0 |
| 34 µm | -0.1 | 1.0 | 2.4 | 12.0 |
| 38 µm | 0.2 | 1.2 | 5.1 | 10.0 |
| 46 µm | -0.1 | 1.5 | 4.5 | 16.0 |



**Table 3. Concentration and liquid water content (LWC) for prescribed droplet distributions calculated with fixed and variable qualified sample area thresholded at 50% actual counts ($SA_{Q\ 50\%}$). Fixed $SA_{Q\ 50\%}$ concentration is not shown because it equals 100 cm$^{-3}$ for all distributions. Uncertainty is equal to 1/2 the range of each parameter when calculated with $SA_{Q\ 10\%}$ and $SA_{Q\ 90\%}$.**

| Distribution Mean [μm] | Variable $SA_{50\%}$ Conc. [cm$^{-3}$] | Variable $SA_{50\%}$ LWC [g m$^{-3}$] | Fixed $SA_{50\%}$ LWC [g m$^{-3}$] | % Diff Variable $SA_{50\%}$ LWC vs. Fixed $SA_{50\%}$ LWC |
|---|---|---|---|---|
| 10 | 106 ± 3 | 0.082 ± 0.002 | 0.079 ± 0.001 | 4.21 |
| 25 | 99 ± 2 | 0.925 ± 0.013 | 0.943 ± 0.016 | -1.84 |
| 35 | 98 ± 1 | 2.294 ± 0.027 | 2.338 ± 0.039 | -1.87 |





**Figures**

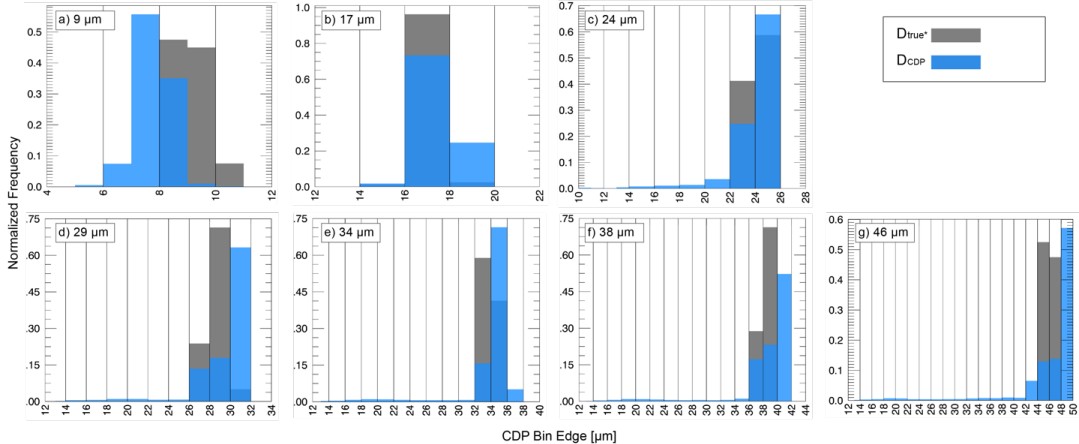

5    **Fig. 1. Normalized distributions of droplet diameter from 80 random glares rounded to the geometric mean of CDP size bins ($D_{true}$*) in grey and CDP-recorded diameter ($D_{CDP}$) from all responses during each test in blue. Nominal droplet size used for each test is indicated in upper left corner.**



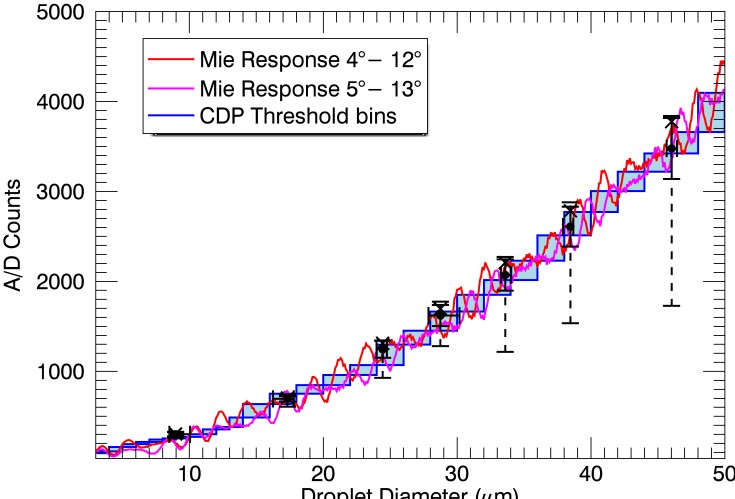

**Fig. 2.** Mie response scaled to CDP A/D counts computed for 4°-12° collection angles (red) and 5°-13° collection angles overlaid on

5    the CDP A/D threshold (shaded blue) that is used to bin individual drops. Black dots show mean droplet diameter ($D_{true}$) for the 7

droplet generator tests. The horizontal bar with end caps represents the 5[th] to 95[th] percentile range of $D_{true}$ for each test. The solid

(dashed) vertical bar with end caps shows the range of the 25[th] to 75[th] (5[th] to 95[th]) percentile CDP-measured A/D counts for each

test. The vertical location of the black dots (X's) show the mean (median) A/D counts for each test.





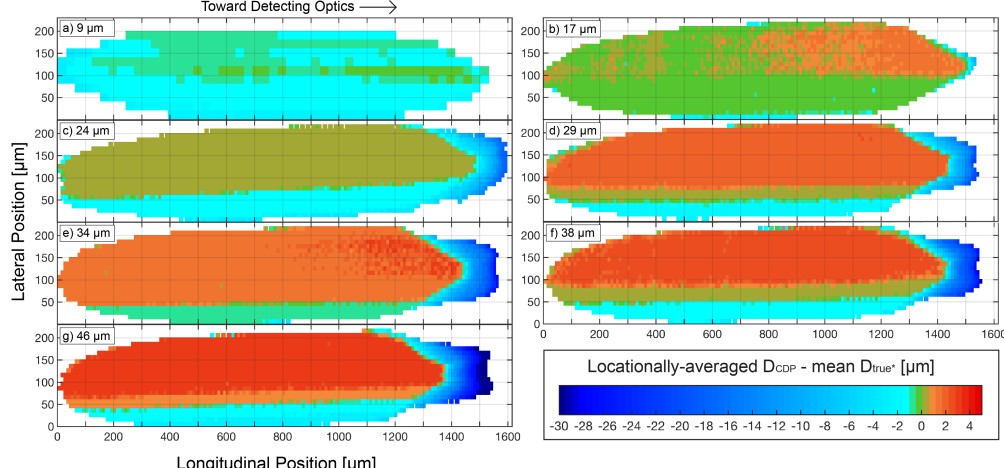

**Fig. 3.** **Beam maps of spatially-dependent sizing accuracy. Colours represent the difference between CDP diameter ($D_{CDP}$)**
5 **averaged at each sample location and mean glare diameter rounded to the geometric mean of CDP size bins ($D_{true}$*) from the 80**
**randomly-selected glares. Droplet diameter used for each map is listed in the upper-left corner. The right side of each map is**
**nearest the detecting optics of the CDP.**




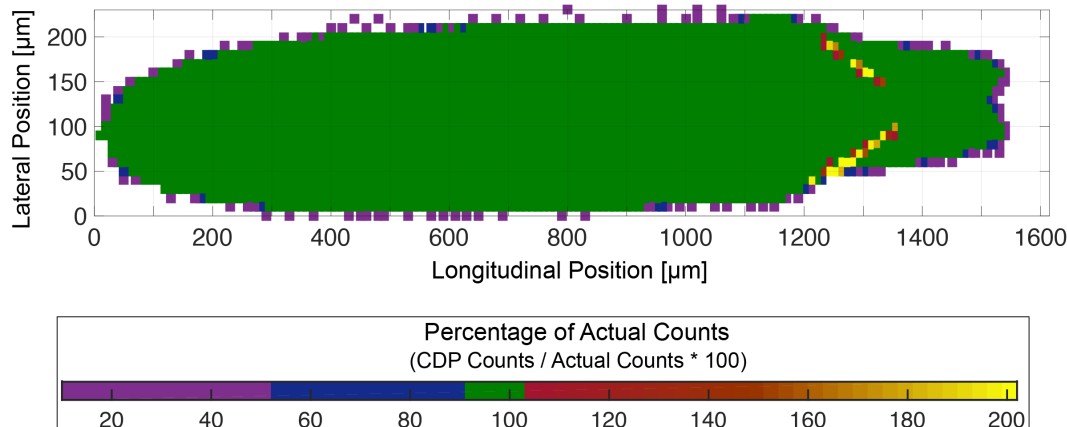

**Fig. 4. Percentage of CDP-recorded counts vs. actual counts (from print head ejection frequency). Purple areas show where at least 10% of actual counts were reported, blue shows where at least 50% were reported, and green shows where at least 90% were reported. Warm colours show areas that received more than 100% of actual counts.**





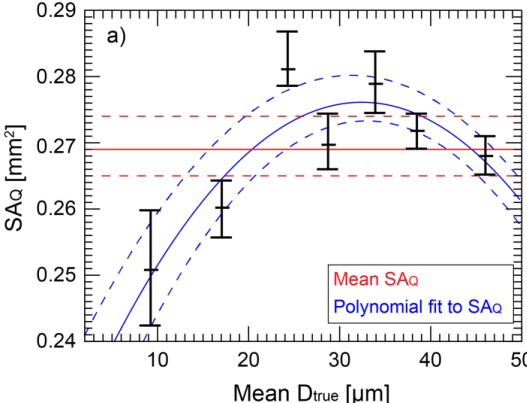

**Fig. 5. Measurements of qualified sample area (SA$_Q$). Horizontal bars represent SA$_Q$ calculated using 10, 50, and 90% true count**
5   **thresholds. Red lines show mean SA$_Q$ for all droplet generator test and blue show a second degree polynomial fit to SA$_Q$.**





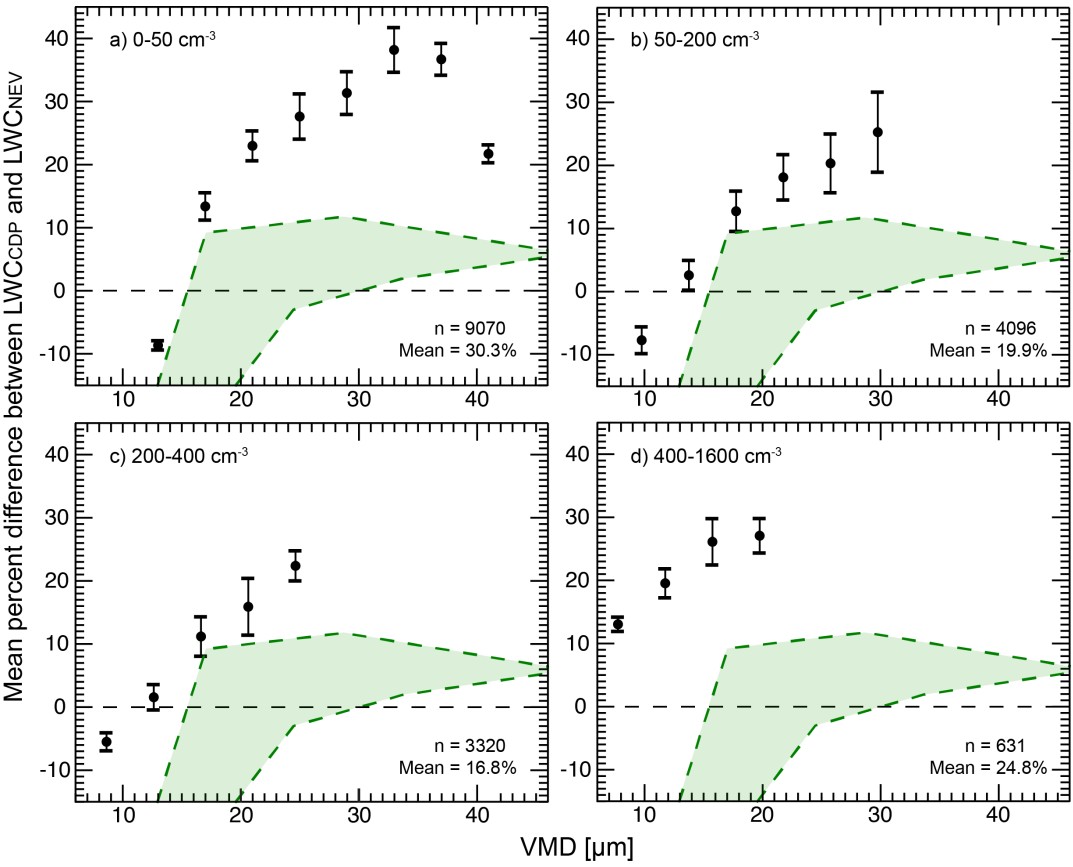

**Fig. 6. Mean percent difference between CDP LWC (LWC$_{CDP}$) and Nevzorov LWC (LWC$_{NEV}$) binned by volume-weighted mean diameter (VMD) for four concentration ranges (shown in upper left corner). Mean percent difference is calculated using a linear regression with the intercept forced through the origin. Error bars show root mean square error. Green dashed areas are estimates of percent difference based on droplet generator tests and Nevzorov collection efficiency considerations. The mean percent LWC difference for all data included in each concentration range and the number of data considered (n) are shown in each plot.**