# Peer review of "Laboratory and in-flight evaluation of measurement uncertainties from a commercial Cloud Droplet Probe (CDP)"

_Atmospheric Measurement Techniques, 2017_

## Referee Comment (RC1) · D. Baumgardner (Referee) · 23 Feb 2018

Manuscript Review

Laboratory and In-flight Evaluation of a Cloud Droplet Probe (CDP)
Spencer Faber, Jeffrey R. French, Robert Jackson

**Overview**

This study of the Cloud Droplet Probe (CDP) is a follow-up to an earlier study by Lance et al. (2010) who established the technique that is used in the present study. Using a droplet generator on a micro-positioner, the authors repeat the measurements published in the earlier Lance et al. paper. As far as I can determine, the only difference in the two studies is that the current study uses a computer controlled positioner, the number of droplet sizes used is larger, covering almost the entire range of the CDP, and a different CDP serial number was used. There do not appear to be any results that contradict those in the earlier study, nor are there any new results that would suggest the need for any serious correction procedures. Hence, I would label this a confirmatory study.

**Major comments**

**Uncertainty in collection angles**

As I state in the overview, it doesn't appear that the current study differs in any significant way from that of Lance et al. If I err in this conclusion, then I suggest that the authors be more clear in the abstract, introduction and summary with respect to how this study distinguishes itself from Lance et al.  I am a firm believer in confirmatory experiments even though they are rarely published. Even though it appears to me that the results support those of Lance et al., the broader size range used in this study justify its publication.

There are several points that are either missing or understated in this paper. Although the authors allude to our Chapter 9 in the AMS monograph, I don't think that they fully evolve the discussion of how important the collection angles are in introducing variations in the derived sizes from the scattering signal. The sizing accuracy for single particle light scattering spectrometers has been estimated as 20% and the concentration accuracy as 16% (Baumgardner, 1983; Dye and Baumgardner, 1984) and every publication since these have used these numbers, including the most recent book on aircraft measurements and our Chapter 9 in the monograph.

The figure from our Chapter 9, shown below, illustrates clearly the issue. With only a $0.5^0$ change in the outer collection angle, there can be differences as much as ±4 um, depending on the size range and whether the actual collection angle is larger or smaller than the nominal. The length of the sample area, i.e. the DOF, is about 1.5 mm. Given that the particle distance from the dump spot determines the collection angle, ±0.75mm changes the amount of scattered light collected and contributes to this uncertainty, particularly for the larger droplets, consistent with the observations.

I am frankly quite surprised that the average differences are so small between the actual versus the measured. These differences are well within the expected uncertainty.

[Figure]

It should be possible to reanalyze the results after recalculating the Mie curves for the correct collection angles and setting new size thresholds to the 30 channels. How to do this? Very straight forward. Just run Mie calculations over a range of angles from perhaps 2 – 15°, in 0.5° steps and fit the results to the droplet measurements, using the particle by particle values that are given in digital counts without pre-binning. The collection angles that produce the best fit are the ones optimum for this particular CDP.

If this unit doesn't have the PbP, the digital scattering values can be interpolated from the channels where the counts fall. With this optimization I predict excellent agreement over all sizes.

**Laser intensity map**

Completely overlooked in this study is the impact of the laser intensity on the sizing and sample area accuracy. This was overlooked in Lance et al. other than a brief mention at the very beginning acknowledging that laser beam intensity gradients can contribute to sizing uncertainty; however, the laser intensity was never mapped in their study.

Given that the intensity distribution is Gaussian across and along the beam, there will be a gradient within the sample area, although the design is meant to minimize missizing by centering the sample area in the flattest intensity region. That being said, without an intensity map that shows how the laser intensity varied within the sample area, it is pure speculation to hypothesize the edge effects on misalignment when some can be explained just be changes in collection angles within the area and others may be do to inhomogeneous laser intensity. This is an issue that can't be dismissed without some serious discussion.

**Summary**

I think that it should be clearly stated that the sizing and sample area uncertainties fall well within those that have been published over the past 25 years. I also think that the

issue of broadening is overstated without and actual estimate of the degree of broadening. Otherwise, it is misleading and possibly even insignificant.

**Minor comments**

Page 1. Line 25: Nominal size range for CDP is 2-50 um.

Page 2, Line 28. Dead-time has not been an issue in the FSSPs since the late 1980's when DMT introduced the SPP-100 replacement electronics for the FSSP.

Page 3, Line 3. "indexes" should be "indices".

Page 4, Line 3, "568" should be "658

Page 4, line 5, "…in an ~12° arc, remove photons in the innermost 5 ~4°, and..". This is poorly worded and confusing. The CDP collects forward scattered light over a solid angle from 4-12$^0$, determined by the distance of the center of focus from the dump spot, the diameter of the dump spot and the aperture in the arm of the CDP.

Page 4, Line 10. "The qualifier's rectangular mask is designed to reduce the collection angles of the detector so that responses are maximized when droplets pass through the qualified sample area.". This is not correct. The mask is designed to accept for sizing only those droplets that pass through the optimum simple area.

---

## Referee Comment (RC2) · Anonymous Referee #2 · 26 Feb 2018

This is a thorough and concise manuscript which I recommend for publication after a couple of changes have been considered. Laboratory and airborne field characterizations of a commercial cloud droplet sizing spectrometer (CDP, Cloud Droplet Probe) are reported. As such it fits well into the scope of Atmos. Meas. Tech.

I have three general and a number of specific comments/suggestions:

General comments

Some discussion about the general applicability of the presented test results of a specific instrument of the CDP type to other instruments of the same type than tested here is required. To which extend the reported sizing deviations are systematic problems of the CDP, and to what extend they hold for the specific probe used here only?

[Figure]

It should be made clearer which progress has been achieved over the results reported previously (in particular with respect to Lance et al., 2010, 2012). Is there more than the incorporation of computerized position stages, instead of manual positioners as used by Lance et al. (2010)?

It should be discussed that the two major parts of the paper (laboratory and field observations) actually don't have much to do with each other. In the lab the sizing was tested, in the field LWC data were evaluated. Also, droplet sizing spectrometers are not made to derive higher order moments of the size distribution, deriving LWC from this type of measurements is originally not intended by these droplet sizers. The immanent sizing uncertainties amplify (cube) in LWC. For LWC measurements different operating principles are much more appropriate, such as the well tested hotwire probes (e.g., the Nevzorov probe). Therefore, it is kind of unfair to compare the extremely error prone LWC data from the droplet sizing spectrometer to the much more straight-forward hotwire bulk probe LWC measurements. Why should you use a droplet spectrometer for LWC measurements if a bulk hotwire instrument is cheaper and much more accurate? I don't say that it makes no sense to compare both LWC measurements, in an ideal world both LWC measurements would agree. However, in the real world I am not surprised at all, that there is a huge discrepancy between the two approaches as illustrated in Figure 6 of the paper. I suggest to include a short discussion on this subject into the manuscript.

Specific suggestions

Title: The wording in the title is inaccurate in the sense that not the CDP probe itself, but its measurement uncertainties in terms of droplet sizing and LWC are evaluated. Maybe the title could be changed to something like "Laboratory and In-flight Evaluation of Uncertainties of Droplet Size and Liquid Water Content Measurements of a Commercial Cloud Droplet Probe". BTW: I always try to avoid acronyms in the title.

Abstract: Please quantify the seven droplets sizes generated in the lab; otherwise it

is unclear what you mean with "For the smallest diameters" (lines 5-6 of abstract) and with "For all larger diameters" (line 7).

1. Introduction

I always try to avoid mentioning specific companies (PMS, DMT) in scientific papers, we should not advertise. That should be replaced by appropriate references to reviewed papers.

I am glad you appreciate the major contribution of Lorenz and call it the "Mie-Lorenz theory" instead of mentioning Mie only. However, to be consequent you should call it "Lorenz-Mie theory", Lorenz was first! And please stay consistent, later in your text you forget Lorenz and use "Mie-theory" only.

I suggest to include the following additional reference when discussing the FAST-FSSP and instrumental broadening of size distribution: Schmidt, S., K. Lehmann, and M. Wendisch, 2004: Minimizing instrumental broadening of the drop size distribution with the M-Fast-FSSP. J. Atmos. Ocean. Tech., 21, 1855–1867.

Page 3, first paragraph: You imply that Lorenz-Mie theory only holds for small droplets, which is not true. It is just not appropriate to apply Lorenz-Mie theory for larger particles; it is much more efficient to use geometric optics for larger particles. But in principle Lorenz-Mie theory is not restricted to small droplets, please would you make respective changes to the text.

2. CDP Operating Principle

A schematic drawing of the CDP would greatly help readers not familiar with this type of instrument to follow your explanation of the principal of operation of the CDP. Alternatively, you may drastically reduce the description of the operational principal and refer to respective literature.

I mostly avoid to refer to gray-literature manuals provided by companies in scientific texts. If unavoidable, just give the web site address.
[Figure]

The differences between FSSP and CDP should be emphasized, progress achieved by CDP in addition to the revised electronics should be highlighted.

Last line in paragraph 20 on page 4: Please quantify "high droplet concentration".

3. University of Wyoming Droplet Generating System

You need to make clear why you partly repeat work done already by Lance et al. (2010).

A schematic illustration of the droplet generator setup might be helpful.

Droplet speed measurements are discussed, at this point the reader asks why this is important. Later it becomes obvious that this is needed to compare with droplet velocities occurring in real aircraft measurements. Just make sure the reader understands here that these measurements have some meaning for the later text.

Maybe you briefly discuss about droplet evaporation during generation and transport to the sampling volume of the CDP.

How about mechanical distortions of the droplet generation setup and its impact on droplet generation, is there a problem with small-scale wind turbulence?

4. Results of Droplet Generator Tests on the CDP

Second paragraph in 4.1: Quantify "Smaller droplets", "shorter test periods", and "less stable".

Figure 1: Maybe you add a similar figure just including droplets passing through the center of depth of field. The axis labels are way too tiny. Coloring is not optimal in my point of view.

I don't see any reason why the pure counting efficiency is not exactly 100 %, please comment on that.

5. Comparison of . . . (please use capital letters for the section title, as you did before)

As discussed above already, you are entering a new world here. Anyway, these data

cannot be compared to your lab studies. Just discuss this, I don't want you to delete this section.

Sometimes you duplicate/repeat figure captions in the text.

The discussion of the differences in Figure 6 is kind of speculative. This is unavoidable, at least partly. A reader could ask why you mostly use data collected in complicated super-cooled conditions with quite some chance to encounter mixed-phase clouds. The matter is already complicated enough in pure liquid water clouds.

The discussion on the droplet speed influence is not satisfying. No way to at least roughly test something in this regard?

———————————————————————

---

## Author Comment (AC1) · 24 Apr 2018

Manuscript Review

Laboratory and In-flight Evaluation of a Cloud Droplet Probe (CDP) Spencer Faber, Jeffrey R. French, Robert Jackson

**Overview**

This study of the Cloud Droplet Probe (CDP) is a follow-up to an earlier study by Lance et al. (2010) who established the technique that is used in the present study. Using a droplet generator on a micro-positioner, the authors repeat the measurements published in the earlier Lance et al. paper. As far as I can determine, the only difference in the two studies is that the current study uses a computer controlled positioner, the number of droplet sizes used is larger, covering almost the entire range of the CDP, and a different CDP serial number was used. There do not appear to be any results that contradict those in the earlier study, nor are there any new results that would suggest the need for any serious correction procedures. Hence, I would label this a confirmatory study.

**We would like to thank Darrel for his thoughtful comments. Below we provide responses to general and specific comments and detail how a revised manuscript will address these issues in an effort towards improving the manuscript.**

**Major comments**

**Uncertainty in collection angles**

As I state in the overview, it doesn't appear that the current study differs in any significant way from that of Lance et al. If I err in this conclusion, then I suggest that the authors be more clear in the abstract, introduction and summary with respect to how this study distinguishes itself from Lance et al. I am a firm believer in confirmatory experiments even though they are rarely published. Even though it appears to me that the results support those of Lance et al., the broader size range used in this study justify its publication.

**It is clear from your comments and those made by other reviewers that more needs to be said about how this study differs from Lance et al.'s 2010 work. In the**

**big picture, the use of computerized vs. manual positioning stages is a minor detail that seems to catch more attention than is warranted.**

**There are a few important ways the droplet generator tests conducted for this work differs from Lance et al.'s 2010 experiments.**
   **· Our experiments tested CDP responses in locations covering the sample area for all seven droplet sizes whereas Lance et al. 2010 performed full "beam maps" for 12 and 22 μm droplets and then tested response at only the center of the sample area using a range of additional droplet sizes. Conducting beam maps for the entire range of droplet sizes provides more information about how droplet size influences the spatially-dependent nature of counting/sizing uncertainty. It also allows for a more complete investigation of how droplet diameter influences sample area dimensions.**
   **· Our experiments used finer spatial resolutions (sample locations every 30 x 20 μm for 9 μm droplets and 10 x 10 μm for all larger droplet sizes) than those conducted by Lance et al 2010 (200 x 20 μm spacing across a majority of the sample area with 50 x 10 μm spacing at the edges of the sample area). Finer resolutions were used to provide a more detailed picture of how CDP performance varies spatially and to provide more precise measurements of sample area dimensions.**
   **· This work was performed on a CDP that includes the pinhole mask modification. Lance et al 2012 did perform droplet generator tests on a modified CDP to measure 'qualified' and 'extended' sample area dimensions but limited detail was provided about the tests, as the paper focused on coincidence error. It seemed worthwhile to conduct detailed droplet generator experiments on a modified CDP to examine how the pinhole mask modification might affect the spatial nature of counting/sizing performance.**

**The revised version of the text will be more explicit about how this study differs from this earlier work.**

There are several points that are either missing or understated in this paper. Although the authors allude to our Chapter 9 in the AMS monograph, I don't think that they fully evolve the discussion of how important the collection angles are in introducing variations in the derived sizes from the scattering signal. The sizing accuracy for single particle light scattering spectrometers has been estimated as 20% and the concentration accuracy as 16% (Baumgardner, 1983; Dye and Baumgardner, 1984) and every publication since these have used these numbers, including the most recent book on aircraft measurements and our Chapter 9 in the monograph.

**There are really two issues that cannot be separated. One, as you correctly point out, is the collection angles. The other is amplitude of the Mie resonances as illustrated in figure 9.2 in Chapter 9 of the Monograph. The latter depends of course on particle size such that expected errors in drop sizing can be largest (~20%) for small particles (d < 10um) and for large particles (d > ~40 um). However, these Mie resonances also depend on the collection angles as illustrated in Figure 2 of the original manuscript (We had a poor choice of color for the two Mie response curves in Fig 2...this makes it hard to distinguish between the response for two different set of collection angles; this will be remedied in a revised manuscript). In the revised manuscript we further explore the impacts of both of these issues and how they relate to each other over a reasonable range of collection angles. This will be discussed in both the introduction and in the context of our measurements, expanding upon the current discussion in section 4.2 at the bottom of page 8 and top of page 9.**

The figure from our Chapter 9, shown below, illustrates clearly the issue. With only a $0.5_0$ change in the outer collection angle, there can be differences as much as ±4 um, depending on the size range and whether the actual collection angle is larger or smaller than the nominal. The length of the sample area, i.e. the DOF, is about 1.5 mm. Given that the particle distance from the dump spot determines the collection angle, ±0.75mm changes the amount of scattered light collected and contributes to this uncertainty, particularly for the larger droplets, consistent with the observations.

**This is an interesting point and one we hadn't really thought of in the context of our measurements. Because of the size of the DOF leading to a change in the collections angles along it, we should expect expect a gradient in sizing. However, our measurements do not show any appreciable change in sizing along the direction. The only test (of the seven performed) that showed any longitudinal dependence on sizing was for 17 um droplet, and based on the individual counts (illustrated by the vertical bar in figure 2) across all of the drops for this test, the gradient in instrument response was very small and likely only shows up on the beam maps (Figure 3) because the probe response was near a threshold size between two bins.**

[Figure]

I am frankly quite surprised that the average differences are so small between the actual versus the measured. These differences are well within the expected uncertainty.

**Quite honestly, so were we. For all of the tests except the 9 um droplets, mean differences were within 1 um and errors based on these averages were on the order of 1-2% in most cases! When we consider the difference between median diameter from the CDP compared to true droplet size, the comparison looks less good, but still well within the 20% expected uncertainty, and mostly within 10%. We do believe this difference (between whether using mean or using median {or modal}) should be considered carefully by investigators when considering calibration for a particular probe. A similar recommendation is suggested in Dye and Baumgardner (1984), but in a different context--they considered issues associated with 'clumping' of calibration beads. For the CDP this needs to be considered because of the skewing of the distribution that results from a severe undersizing of a very small percentage of droplets, at the far end of the DOF.**

It should be possible to reanalyze the results after recalculating the Mie curves for the correct collection angles and setting new size thresholds to the 30 channels. How to do this? Very straight forward. Just run Mie calculations over a range of angles from perhaps $2 - 15_o$, in $0.5_0$ steps and fit the results to the droplet measurements, using the particle by particle values that are given in digital counts without pre-binning. The collection angles that produce the best fit are the ones optimum for this particular CDP. If this unit doesn't have the PbP, the digital scattering values can be interpolated from the channels where the counts fall. With this optimization I predict excellent agreement over all sizes.

**We began to investigate something very similar to what you describe here and actually reached a conclusion that it was anything but straightforward. In Figure 2, we provide Mie Response curves for two sets of collection angles (4-12 and 5-13). We overlay on these curves results from the PBP output for each of the tests. In this example, it is not clear to us which set of curves best represents the data.**

**Part of the difficulty comes down to determining the proper scaling between the Mie Calculations and the A/D counts output by the probe. The actual value of this scaling factor is not known, such that each individual curve is scaled to fit within the A/D range of the probe. IF we knew the actual scaling factor, which must be a function of the gain and the relationship between incident light intensity and probe voltage output, the true scale factor could be determined. Discussions with engineers at DMT indicated they used a methodology similar to what we describe above.**

**Laser intensity map**

Completely overlooked in this study is the impact of the laser intensity on the sizing and sample area accuracy. This was overlooked in Lance et al. other than a brief mention at the very beginning acknowledging that laser beam intensity gradients can contribute to sizing uncertainty; however, the laser intensity was never mapped in their study.

Given that the intensity distribution is Gaussian across and along the beam, there will be a gradient within the sample area, although the design is meant to minimize missizing by centering the sample area in the flattest intensity region. That being said, without an intensity map that shows how the laser intensity varied within the sample area, it is pure speculation to hypothesize the edge effects on misalignment when some can be explained just be changes in collection angles within the area and others may be do to inhomogeneous laser intensity. This is an issue that can't be dismissed without some serious discussion.

**We agree and appreciate you bringing this point to us. Because we don't measure the laser intensity, we cannot comment directly on its impact but it does indeed provide a plausible explanation for our results showing lateral-dependent sizing. Within the revised manuscript we provide more thorough discussion following the (modified) first paragraph in the Section 6 (renamed Summary and Discussion).**

**This expanded discussion explores all three potential contributors to missizing: laser inhomogeneities, optical misalignment, and Mie resonances & scattering angles. The 'flavors' of missizing reported in the manuscript (general over/under sizing for a particular droplet diameter vs lateral dependence) may likely come from different sources due to nature of the physical manifestations of these three potential contributors. This is also discussed in the revised manuscript along with specific recommendations for future studies.**

**Summary**

I think that it should be clearly stated that the sizing and sample area uncertainties fall well within those that have been published over the past 25 years. I also think that the issue of broadening is overstated without and actual estimate of the degree of broadening. Otherwise, it is misleading and possibly even insignificant.

**In the revised manuscript we clearly state that the uncertainties fall within the expected range reported in earlier studies.**

**We also use the PBP measurements to quantify the broadening based on the width of the 'true' droplet sizes and the width of the measured droplets for each test. These data will be included in one of the tables (likely table 2) or added as an additional table in the manuscript.**

**Minor comments**

Page 1. Line 25: Nominal size range for CDP is 2-50 um.
**The nominal size range of the CDP will be corrected in the revision.**

Page 2, Line 28. Dead-time has not been an issue in the FSSPs since the late 1980's when DMT introduced the SPP-100 replacement electronics for the FSSP.
**The introduction will include discussion about how the SPP-100 electronics negate dead time losses in FSSP measurements.**

Page 3, Line 3. "indexes" should be "indices".
**This will be corrected in the revised version.**

Page 4, Line 3, "568" should be "658
**Thank you for catching that mistake. It will be corrected.**

Page 4, line 5, "...in an ~12° arc, remove photons in the innermost 5 ~4°, and..". This is poorly worded and confusing. The CDP collects forward scattered light over a solid angle from 4-12$_0$, determined by the distance of the center of focus from the dump spot, the diameter of the dump spot and the aperture in the arm of the CDP.

**The wording has been corrected.**

Page 4, Line 10. "The qualifier's rectangular mask is designed to reduce the collection angles of the detector so that responses are maximized when droplets pass through the qualified sample area.". This is not correct. The mask is designed to accept for sizing only those droplets that pass through the optimum simple area.

**The wording has been corrected.**

---

## Author Comment (AC2) · 24 Apr 2018

This is a thorough and concise manuscript which I recommend for publication after a couple of changes have been considered. Laboratory and airborne field characterizations of a commercial cloud droplet sizing spectrometer (CDP, Cloud Droplet Probe) are reported. As such it fits well into the scope of Atmos. Meas. Tech.

I have three general and a number of specific comments/suggestions:

**We appreciate the feedback from the anonymous reviewer. Below, inline, we provide responses to the reviewer's general and specific comments.**

General comments

Some discussion about the general applicability of the presented test results of a specific instrument of the CDP type to other instruments of the same type than tested here is required. To which extend the reported sizing deviations are systematic problems of the CDP, and to what extend they hold for the specific probe used here only?

**We strongly agree with the sentiment that 'to what extent our findings, for this specific probe, are more generally applicable to other CDPs' is an important question'. In the original manuscript, we briefly mention at the end of paragraph 1 in the summary that the lateral dependence of sizing was also measured in two other CDPs tested on our system. It turns out, that those measurements also indicate a 1-2 um oversizing through much of the sampled region for drops larger than about 25 um. We will add the results of these measurements to the summary in the revised manuscript. However, to some extent, these measurements are anecdotal only. But they do show at least some consistency across different CDP instruments.**

**We also note in the original summary that a similar lateral dependence on sizing was reported by Lance et al (2010); although (and not discussed in our original manuscript) their limited resolution and test sizes did not reveal how this dependence changed with droplet size. The position dependent-sizing described by Lance et al (2010) shown in their figure 6a & 6d differed from our results. In particular, where it occurred within the sample volume--in all of our tests, the mis-sizing of droplets was rather symmetric with a lateral dependence across the sample volume (laser beam), and a small region near the detector where droplets were severely undersized. It may be that the specific probe described by Lance et al was not well-aligned (as suggested in their paper), and for later generations of the CDP, the optical alignment is better positioned/characterized by the manufacturer. Regardless, the missizing, in response to sample location, appears to have much better symmetry with the sample volume than reported by Lance. This discussion is added to the summary section of the revised manuscript (now revised as Section 6: Summary and Discussion)**

It should be made clearer which progress has been achieved over the results reported previously (in particular with respect to Lance et al., 2010, 2012). Is there more than the incorporation of computerized position stages, instead of manual positioners as used by Lance et al. (2010)?

**We discuss some of this above in our response to general comment #1. In terms of the droplet generator system, yes, only the addition of computerized stages has progressed over the system originally designed and described by Lance et al (2010). However, this manuscript is meant to focus not on the droplet generating**

**system, but rather how using that system enables further evaluation of CDP performance. To that end, the UW droplet generator provides a more thorough set of the experiments covering a broader range of droplets with higher spatial resolution across the sample volume. Lance et al.'s 2010 work tested CDP response at a grid of location throughout the sample area using only 12 and 22 μm droplets. This study includes similar experiments for seven droplet sizes, conducted at a much finer spatial resolution.**

**The revised manuscript elucidates how these additions provide a greater understanding of instrument response across its entire sample volume over the range of particles it was designed to measure. Much of this addition is in the summary and discussion in the revised manuscript. We also lay the groundwork for this in the introduction (Section 1) and discussion of the UW droplet generating system (Section 3).**

It should be discussed that the two major parts of the paper (laboratory and field observations) actually don't have much to do with each other. In the lab the sizing was tested, in the field LWC data were evaluated. Also, droplet sizing spectrometers are not made to derive higher order moments of the size distribution, deriving LWC from this type of measurements is originally not intended by these droplet sizers. The immanent sizing uncertainties amplify (cube) in LWC. For LWC measurements different operating principles are much more appropriate, such as the well tested hotwire probes (e.g., the Nevzorov probe). Therefore, it is kind of unfair to compare the extremely error prone LWC data from the droplet sizing spectrometer to the much more straight-forward hotwire bulk probe LWC measurements. Why should you use a droplet spectrometer for LWC measurements if a bulk hotwire instrument is cheaper and much more accurate? I don't say that it makes no sense to compare both LWC measurements, in an ideal world both LWC measurements would agree. However, in the real world I am not surprised at all, that there is a huge discrepancy between the two approaches as illustrated in Figure 6 of the paper. I suggest to include a short discussion on this subject into the manuscript.

**In terms of cloud property measurements, closure studies are very important. One way we accomplish this is by comparing different types of measurements--for example, those that respond to different aspects of the particle size distribution such as the 2nd moment (effective radius), 3rd moment (mass), or 6th moment (Rayleigh reflectivity). The point of all of this is that under the appropriate conditions, these closure tests should agree to within the uncertainty of the individual measurements after allowing for error propagation. The fact that we do not find complete agreement between the bulk LWC and the CDP-derived measurements is not a surprise (as you state). But rather, that the disagreement between the two is greater than what is able to be quantified by our best estimate of uncertainty in *both* measurements is disconcerting. This tells us that our error quantification for one (or both) of the measurements is incorrect. In the revised manuscript we add discussion in this regard. For the case presented here, one must accept uncertainty in LWC greater than that which is suggested based strictly on the Hotwire (or CDP) uncertainty estimates alone resulting from this closure argument. In the revised manuscript, we add discussion to this regard in section 5.4.**

Specific suggestions

Title: The wording in the title is inaccurate in the sense that not the CDP probe itself, but its measurement uncertainties in terms of droplet sizing and LWC are evaluated. Maybe the title could be changed to something like "Laboratory and In-flight Evaluation of Uncertainties of Droplet Size and Liquid Water Content Measurements of a Commercial Cloud Droplet Probe". BTW: I always try to avoid acronyms in the title.

**We have changed the title in the revised manuscript to: "Laboratory and In-flight Evaluation of Uncertainties of Measurements from a Cloud Droplet Probe (CDP)"**

**We acknowledge that the measurements are being evaluated and we spell-out the name of the probe. However, we suggest maintaining CDP within a parenthetical because it is well-established in the airborne cloud physics community (such as FSSP).**

Abstract: Please quantify the seven droplets sizes generated in the lab; otherwise it is unclear what you mean with "For the smallest diameters" (lines 5-6 of abstract) and with "For all larger diameters" (line 7).

**In the revised abstract we are explicit about the drop sizes tested. We state: "...for 9 um drops the CDP undersized drops by 1 - 4 um…Droplets of 17 to 24 um were sized to within 2 um of the correct diameter… ...For droplets 24 um and larger …"**

1. Introduction

I always try to avoid mentioning specific companies (PMS, DMT) in scientific papers, we should not advertise. That should be replaced by appropriate references to reviewed papers.

**Fair enough. While this is difficult in such a small field and (because of that) we often have probes with similar acronyms (for instance the DMT CDP and the SPEC F-CDP which are two different probes)...we have revised the manuscript to remove manufacturer's name wherever possible. However, in some instance, no other references to certain probes exist and we provide links to online instrument manuals.**

I am glad you appreciate the major contribution of Lorenz and call it the "Mie-Lorenz theory" instead of mentioning Mie only. However, to be consequent you should call it "Lorenz-Mie theory", Lorenz was first! And please stay consistent, later in your text you forget Lorenz and use "Mie-theory" only.

**Instances of "Mie" have been standardized to "Lorenz-Mie" throughout the document.**

I suggest to include the following additional reference when discussing the FAST-FSSP and instrumental broadening of size distribution: Schmidt, S., K. Lehmann, and M. Wendisch, 2004: Minimizing instrumental broadening of the drop size distribution with the M-Fast-FSSP. J. Atmos. Ocean. Tech., 21, 1855–1867.

**This reference will be included in the discussion of OPC sizing uncertainty sources.**

Page 3, first paragraph: You imply that Lorenz-Mie theory only holds for small droplets, which is not true. It is just not appropriate to apply Lorenz-Mie theory for larger particles; it is much more efficient to use geometric optics for larger particles. But in principle Lorenz-Mie theory is not restricted to small droplets, please would you make respective changes to the text.

**Of course, you are correct...all scattering follows Lorenz-Mie theory. In the revised manuscript we state: 'due to the relationship between particle size (d~10-50 um) and wavelength of incident radiation (lambda~658 nm), full Lorenz-Mie theory calculations must be considered to accurately relate droplet size to scattered intensity.**

2. CDP Operating Principle

A schematic drawing of the CDP would greatly help readers not familiar with this type of instrument to follow your explanation of the principal of operation of the CDP. Alternatively, you may drastically reduce the description of the operational principal and refer to respective literature.

I mostly avoid to refer to gray-literature manuals provided by companies in scientific texts. If unavoidable, just give the web site address.

**We chose not to include a schematic of the CDP because this is included in both Lance et al. (2010) and the DMT manual (2014). However, we felt a brief description was appropriate here with the appropriate references at the end of the paragraph.**

**In principle, we agree with your comments regarding 'gray literature'. However, very little is available in the peer-reviewed literature regarding 'relatively' new instruments such as the CDP. Perhaps, even more important, is that in-production probes are constantly changing. By capturing manufactures' snapshots of designs at a specific time (based on a specific manual revision, 2014 in this case) we can help alleviate confusion resulting from comparisons of differing probe generations.**

The differences between FSSP and CDP should be emphasized, progress achieved by CDP in addition to the revised electronics should be highlighted.

**The revised version of the manuscript will include discussion about other improvements made to the CDP (in addition to revised electronics and the exclusion of the laser shroud) including the unimodal laser intended to provide more homogenous laser intensity within the sample area and the differences in qualifier mask design.**

Last line in paragraph 20 on page 4: Please quantify "high droplet concentration".

**Standard coincidence depends upon flight speed (faster speed, greater probability for two particles being in the sample volume at same time). But at most research aircraft flight speeds (~80-120 m s-1), standard coincidence is anticipated at less than 5% for concentrations on the order of 500 cm-3 based on a qualified sample volume of 0.3 mm2 (Lance et al. 2010).**

3. University of Wyoming Droplet Generating System
You need to make clear why you partly repeat work done already by Lance et al. (2010). A schematic illustration of the droplet generator setup might be helpful.

**As noted above (and elucidated in the revised manuscript), the focus of this work is not to expand on the droplet generator setup (from Lance et al. 2010) but rather to expand on the measurements they provide. The revised manuscript describes how the UW setup allows a greater range of testing, across the full dynamic range of droplet sizes measured by the CDP and the full volume sampled.**

Droplet speed measurements are discussed, at this point the reader asks why this is important. Later it becomes obvious that this is needed to compare with droplet velocities occurring in real aircraft measurements. Just make sure the reader understands here that these measurements have some meaning for the later text.

**In the revised manuscript, we add text to describe to the reader that velocity measurements are needed to ensure that droplets are traveling at speeds within the operational airspeed of the CDP and to compare droplet velocity with typical aircraft airspeeds.**

Maybe you briefly discuss about droplet evaporation during generation and transport to the sampling volume of the CDP.
How about mechanical distortions of the droplet generation setup and its impact on droplet generation, is there a problem with small-scale wind turbulence?

**We, in part, control droplet size through evaporation. Increased residence time within the sheath flow increases evaporation and allows us to tune our experiment to a specific drop size (i.e. increasing residence time in the sheath flow reduces droplet diameter...therefore we can increase/decrease diameter by ejecting the droplets later/earlier in the flow field). The range of residence time in the sheath flow is provided in the revised manuscript.**

**Turbulence in the vicinity of the flow tube exit is an important consideration and is the main factor limiting droplet velocity. It turns out that turbulence at the point of droplet generation does not appear to be important (based on high speed photography of droplet generation events). However, the acceleration of those droplets (once generated) may induce distortions in the droplets as they are ejected from the accelerated flow into the CDP sample volume. Because we measure (independently) the size of droplets as they pass through the sample volume of the CDP (through the glare technique) we assume any impact of distortion is accounted for; however departures from spheroidal shape may result in slight mis-sizing...this is acknowledged in the revised manuscript; although the data collected does not allow for quantification.**

4. Results of Droplet Generator Tests on the CDP

Second paragraph in 4.1: Quantify "Smaller droplets", "shorter test periods", and "less stable".

**In the revised manuscript we better describe the constraints of the experimental setup. In particular, for the smallest droplet produced (9 um) we needed to complete the test runs in roughly 25% of the time required for larger droplets. This was because we found that after a certain length of time, the size of the droplets (and/or the position of the droplets) was no longer predictable. Because of the shorter test periods, the resolution of the measurements was reduced and the number of droplets at each location was also reduced. The same was true for the 17 um resolution test, although the test was conducted over roughly 75% of the time required for larger droplets; so that the statistics are significantly improved. All other tests (for drops 24 um and larger) were completed at the same temporal and spatial resolution.**

Figure 1: Maybe you add a similar figure just including droplets passing through the center of depth of field. The axis labels are way too tiny. Coloring is not optimal in my point of view.

**The font size of axis labels for Figure 1 will be increased for better readability. The semi-transparent bars in Figure 1 are blue and a neutral grey so that overlapping regions don't appear as a third color. Changing the grey bars to a more obvious color could complicates interpretation of the figure.**

**Center of DOF measurements can be quite misleading. While this may (or may not) represent how the manufacturer calibrates the CDP, it does not represent what the probe itself measures in a cloud. In fact, we**

**assert that the range of droplet mis-sizing quantified in Figures 1 and 3 is key to understanding how best to calibrate the probe. As we discuss in the summary, determining whether to calibrate based on particle modal diameter or mean diameter has impacts across the droplet spectrum. Because the degree of mis-sizing depends on the droplet location within the sample volume, we believe that plotting the droplet size within a specific location (such as the center of DOF) is misleading and can result in wrong conclusions. For this, we maintain, that describing the response of the probe throughout the entire sample volume using a (nearly) monodisperse distribution is the best way to represent measurement sizing accuracy.**

I don't see any reason why the pure counting efficiency is not exactly 100 %, please comment on that.

**In general, we agree. Of course, on the very edges (within one 10 X 10 um sample bin) a droplet might fall on the edge resulting in reduced counts...but the more curious points in our study result from the over counting in the small region near the detector. Our results suggest something is occurring that is not well accounted for in the model of how the probe responds. We have no suggestion as to what might cause this result. While the overall effect (impact on total number counted) is small, it is curious that it occurs in the region located between the 'expected' normal sample volume and the 'region outside of the expected sample volume, where particles are severely under-sized'. The revised manuscript discusses this in sections 4 and 6, but at this time we don't offer any concrete explanations.**

**We find it curious that Lance et al (2010) found significantly larger mis-counting at different locations for their two tests (Figs 6b & 6e). Their results appear much more difficult to reconcile than our findings. This is discussed in the revised manuscript.**

5. Comparison of . . . (please use capital letters for the section title, as you did before)

**Some words in the section title are not capitalized in order to follow AMT guidelines: 'titles and headings should follow sentence-style capitalization standards (only first words and proper nouns are capitalized)'. Section titles with unnecessary capitalization will be corrected.**

As discussed above already, you are entering a new world here. Anyway, these data cannot be compared to your lab studies. Just discuss this, I don't want you to delete this section.

**Agreed, our discussion above (and included in the revised manuscript) makes this point.**

Sometimes you duplicate/repeat figure captions in the text.

**The revised manuscript removes these duplications.**

The discussion of the differences in Figure 6 is kind of speculative. This is unavoidable, at least partly. A reader could ask why you mostly use data collected in complicated super-cooled conditions with quite some chance to encounter mixed-phase clouds. The matter is already complicated enough in pure liquid water clouds.

**Discussion of Figure 6 is quite speculative. Perhaps the main takeaway from this section is that further work using different data and additional lab experiments is needed to get a better understanding of exactly what is leading to the greater than expected difference between LWC estimates provided by the two instruments.**

**A majority of  data used in this analysis were taken in super-cooled conditions because there were very few cloud penetrations with temperatures greater than 0 C encountered during SNOWIE and PACMICE. Comparing measurements from the two instruments in conditions that could contain liquid and ice particles certainly isn't ideal but IWC estimates from the Nevzorov were used to exclude periods that contained a significant number of ice particles. There are surely some mixed phase penetrations still included in the analysis but we believe ice particles minimally affected results because the Nevzorov LWC sensor has been shown to only be sensitive to 12% IWC (as demonstrated by Korolev et al., 1998), with IWC typically being small compared to LWC in mixed phase conditions. Also, Lance et al (2010) and Khanal (2018) have shown that the CDP is not greatly affected by natural or shattered ice particles.**

The discussion on the droplet speed influence is not satisfying. No way to at least roughly test something in this regard?

**Ideally, additional droplet generator tests using greater velocities would have been conducted to test how droplet speed influences the results. This research was conducted to satisfy requirements for a masters degree so there unfortunately was not time to make necessary modifications to the laboratory setup and run additional experiments.**